# Alcohol effects on globus pallidus connectivity: Role of impulsivity and binge drinking

Samantha J. Fede[1], Karina P. Abrahao[2,3], Carlos R. Cortes[1], Erica N. Grodin[1], Melanie L. Schwandt[4], David T. George[4], Nancy Diazgranados[4], Vijay A. Ramchandani[5], David M. Lovinger[2], Reza Momenan[1]*

**1** Clinical NeuroImaging Research Core, National Institute on Alcohol Abuse and Alcoholism, National Institutes of Health, Bethesda, Maryland, United States of America, **2** Laboratory for Integrative Neuroscience, National Institute on Alcohol Abuse and Alcoholism, National Institutes of Health, Rockville, Maryland, United States of America, **3** Departamento de Psicobiologia, Universidade Federal de Sao Paulo, Sao Paulo, São Paulo, Brazil, **4** Office of the Clinical Director, National Institute on Alcohol Abuse and Alcoholism, National Institutes of Health, Bethesda, Maryland, United States of America, **5** Section on Human Psychopharmacology, National Institute on Alcohol Abuse and Alcoholism, National Institutes of Health, Bethesda, Maryland, United States of America

☯ These authors contributed equally to this work.
* reza.momenan@nih.gov

**Data Availability Statement:** Given requirements of the National Institutes of Health Institutional Review Board regarding protection of human subjects data, and ethical restrictions to data being

## Abstract

Despite the harm caused by binge drinking, the neural mechanisms leading to risky and disinhibited intoxication-related behaviors are not well understood. Evidence suggests that the globus pallidus externus (GPe), a substructure within the basal ganglia, participates in inhibitory control processes, as examined in stop-signaling tasks. In fact, studies in rodents have revealed that alcohol can change GPe activity by decreasing neuronal firing rates, suggesting that the GPe may have a central role in explaining impulsive behaviors and failures of inhibition that occur during binge drinking. In this study, twenty-five healthy volunteers underwent intravenous alcohol infusion to achieve a blood alcohol level of 0.08 g/dl, which is equivalent to a binge drinking episode. A resting state functional magnetic resonance imaging scan was collected prior to the infusion and at binge-level exposure. Functional connectivity analysis was used to investigate the association between alcohol-induced changes in GPe connectivity, drinking behaviors, and impulsivity traits. We found that individuals with greater number of drinks or heavy drinking days in the recent past had greater alcohol-induced deficits in GPe connectivity, particularly to the striatum. Our data also indicated an association between impulsivity and alcohol-induced deficits in GPe—frontal/precentral connectivity. Moreover, alcohol induced changes in GPe-amygdala circuitry suggested greater vulnerabilities to stress-related drinking in some individuals. Taken together, these findings suggest that alcohol may interact with impulsive personality traits and drinking patterns to drive alterations in GPe circuitry associated with behavioral inhibition, possibly indicating a neural mechanism by which binge drinking could lead to impulsive behaviors.

available publicly, such as the data being potentially identifying or a lack of patient consent to publicly share the data, we will make the data available to qualified researchers upon request. We have identified a non-author contact point who will process all data requests and provide access to de-identified data as appropriate: Mike Kerich, mikek@nih.gov.

**Funding:** This research was supported by the National Institutes of Health (NIH) intramural funding ZIA AA000061-22 LCTS (Clinical NeuroImaging Research Core; to RM) and AA000416 (Corticostriatal Mechanisms of Action Learning and Habit Formation; to DML), Division of Intramural Clinical and Biological Research of the National Institute on Alcohol Abuse and Alcoholism (NIAAA). The funders had no role in study design, data collection and analysis, decision to publish, or preparation of the manuscript.

**Competing interests:** The authors have declared that no competing interests exist.

## Introduction

Harmful drinking behaviors like binge drinking contribute to 5.9% of all global deaths [1, 2]. This is largely due to the development of alcohol use disorder (AUD) and consequential organ damage caused by chronic drinking, intoxication-related car accidents and domestic violence [3, 4]. Binge drinking is defined as drinking that results in blood alcohol concentration (BAC) levels at or above 0.08 g/dl (estimated as consuming at least 4 or 5 standard drinks in less than two hours for women and men, respectively). The mechanisms underlying binge drinking and related risky behaviors are unclear, but impulsivity or failure in inhibitory control during decision-making likely plays a role [5–8].

Research into the neural correlates of failure to inhibit suggests that the basal ganglia plays an important role [9, 10]. The basal ganglia are formed by several interconnected brain regions that control action selection, reward, goal-directed behavior and habitual learning [11]. Frontostriatal pathways enable inhibitory control quickly via a hyperdirect pathway through the inferior frontal gyrus (IFG) and presupplementary motor area (preSMA) that quickly "brakes", and through indirect pathways that "suppresses" movement [12, 13]. In particular, the globus pallidus externus (GPe), a central and highly interconnected component of the basal ganglia, may have a key role in inhibitory control [14]. A specific projection from the GPe to the striatum known as the arkypallidal projection has been shown to inhibit behavior temporarily in "stop-signal"-based tasks [15, 16]. Given this role, alterations to GPe connectivity could contribute to increased impulsivity and failure of inhibitory control.

Alcohol has been shown to have a direct effect on GPe neurons. In a mouse model, Abrahao et al. [17] have demonstrated that acute alcohol decreases the firing rate of specific neurons in the GPe, including (1) the low frequency prototypical neurons, which project downstream to the subthalamic nucleus / substantia nigra and upstream to the striatum, and (2) the arkypallidal neurons, which project to the striatum. In humans, alcohol decreased functional magnetic resonance imaging (fMRI) signal in the GP during a stop-signal task [18], and this signal change was associated with slowed reaction time and impaired inhibitory control [10]. Taken together, these findings suggest that impulsivity and behavioral disinhibition associated with alcohol consumption may be driven by alcohol-induced changes in GPe connectivity.

In this study, we aimed to extend the preclinical results of the Abrahao et al, 2017 study to a human paradigm by determining whether alcohol-induced changes in GPe connectivity are related to recent drinking patterns, impulsivity, and interactions between the two. To do so, we compared the resting-state fMRI connectivity of the GPe before and after controlled acute intravenous (IV) alcohol administration to "binge drinking" levels (BAC = 0.08 g/dl) in healthy humans, and examined the relationship between those differences, recent drinking history, and trait impulsivity. To our knowledge, this is the first time such an investigation has been conducted. We hypothesized that alcohol would result in greater decreases in GPe functional connectivity across the brain in individuals with more impulsive traits and higher risk drinking behaviors. Moreover, we hypothesized that the interaction between recent drinking behaviors and impulsivity would contribute to alcohol-induced changes in GPe functional connectivity.

## Methods

This study was approved by the NIH Addictions Institutional Review Board under 12-AA-0032. All subjects provided written informed consent prior to their participation in the study.

### Participants

Twenty-five healthy (13 males, 12 females; mean age: 28.85 years), light social drinkers were recruited by local advertisement, according to approved National Institutes of Health

Institutional Review Board procedures. The demographic data of participants is summarized in Table 1. After obtaining written informed consent, all participants underwent a comprehensive medical screen, including blood work, urinalysis, medical history, physical exam, and a Structured Clinical Interview for the DSM Axis-I Disorders (SCID-IV) [19]. Criteria for inclusion in this study were: healthy, 21–45 years old, consumption of 1 to 10 drinks per week for females and 1 to 14 drinks per week for males. Subjects also had to have consumed at least two standard drinks of alcohol within one hour on at least one occasion in their lifetime. Criteria for exclusion included: abnormal blood or urine lab test values or findings from the medical screen, DSM-IV criteria for alcohol or other substance dependence (excluding nicotine) at any time; current or past major psychiatric disorder (DSM-IV Axis I), head injury requiring hospitalization, Body Mass Index (BMI) value over 30, inability to stop taking any medication or drugs 3 days prior to study days, and MRI contraindications. Also, non-drinkers were excluded from the study due to ethical concerns related to alcohol administration.

## Measures

**Alcohol drinking measures.** The amount of daily alcohol consumption over the last 90 days was measured using the Alcohol Timeline-Followback (TLFB) calendar—a drinking assessment method with good psychometric characteristics for estimating retrospective daily drinking patterns [20]. The number of heavy drinking days in the last 90 days was calculated from the TLFB; a heavy drinking day is defined as 4 or more drinks per day for women and 5 or more drinks per day for men. We used this as a measure for binge drinking. Total drinks in the past 30 days was also calculated from the TLFB. Past 30-day drinking was used rather than 90-day drinking given that we were interested in recent drinking, and that recent reports of drinking may be more reliable.

**Table 1. Demographic, drinking, and impulsivity characteristics.**

| Variable | Count | |
|---|---|---|
| Total n | 25 | |
| Gender (Male / Female) | 13 / 12 | |
| Smokers | 1 | |
| | **Mean** | **SD** |
| Age | 28.85 | 7.40 |
| Years Of Education | 15.88 | 1.88 |
| BIS: Total | 54.76 | 9.62 |
| Attentional Impulsiveness | 12.96 | 2.99 |
| Motor Impulsiveness | 20.92 | 3.32 |
| Nonplanning Impulsiveness | 19.72 | 4.21 |
| UPPS-P: Positive Urgency | 1.36 | 0.31 |
| Negative Urgency | 1.66 | 0.44 |
| Premeditation | 1.78 | 0.42 |
| Perseverence | 1.53 | 0.39 |
| Sensation Seeking | 2.85 | 0.70 |
| Delay Discounting Rate (K) | 0.02 | 0.06 |
| TLFB—Drinks Per Thirty Days | 20.19 | 13.69 |
| TLFB—Heavy Drinking Days | 19.08 | 10.96 |

Abbreviations as follow: BIS- Barratt Impulsiveness Scale; TLFB- Alcohol Timeline Followback.

**Impulsivity measures.** Impulsive behavioral traits were measured using the Barratt Impulsiveness Scale (BIS-11) and the UPPS-P Impulsive Behavior Scale, two of the most widely-used self-report tools in the evaluation of trait impulsivity [21]. The BIS-11 scale evaluates impulsive behavior in general, and the motor, attentional and non-planning sub-scales evaluate acting without thinking, inability to focus attention and lack of forethought, respectively. The UPPS-P measures personality traits conducive to impulsive behavior including **U**rgency (negative)—tendency to impulsively act under strong negative emotions, **P**remeditation (lack of)—tendency to act without thinking, **P**erseverance (lack of) to remain focused on a task, **S**ensation-seeking—tendency to seek out novel/thrilling experiences, and **P**ositive-urgency—tendency to impulsively act under strong positive emotions.

We also evaluated choice impulsivity with the Delay Discounting Task (DDT). The DDT [22, 23] measures impulsivity by presenting the subject with a series of hypothetical choices between receiving a smaller immediate monetary reward or a larger delayed reward. Subjects were presented as their immediate reward values between $100 and $0 dollars in increments of $10, while the delayed reward was $100. Delays were 0, 7, 14, 20, 25, or 30 days. Subjects completed trials with all iterations of these parameters. The rate of discounting of the delayed outcome (k) was calculated and then a natural log-transformation was applied to correct for the non-normal distribution of k values. The resulting ln(k) is used to represent delay discounting, where higher ln(k) values mean greater preference for immediate rewards (see Table 1 for descriptive statistics of drinking and impulsivity scores for the sample).

## Experimental design and statistical analysis

**Overall timeline.** In this two-session study, healthy light drinkers received an IV alcohol infusion on two separate days. The first session was conducted outside the scanner to establish the alcohol infusion rate profile to achieve and maintain the target BrAC exposure, and to ensure tolerability. The alcohol infusion for the second session was conducted inside the scanner at least three days after the first session (Fig 1). Participants were asked to fast the night

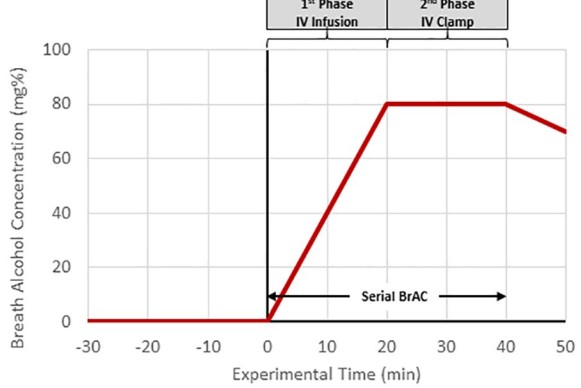
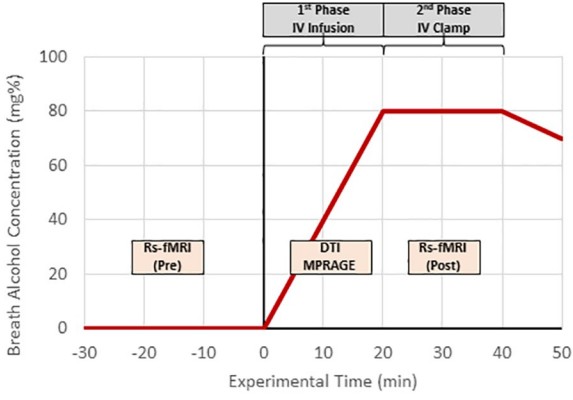

**Fig 1. Timeline of intravenous alcohol infusion study.** The timeline depiction of the two-session intravenous alcohol infusion study; sessions were separated at least three days from each other. The session 2 timeline indicates timepoints of resting state and BAC measurement alongside IV infusion; the first resting state scan occurred before the IV infusion at blood alcohol level of 0.0 g/dl and the second one was collected when the blood alcohol level reached the binge levels (0.08 g/dl).

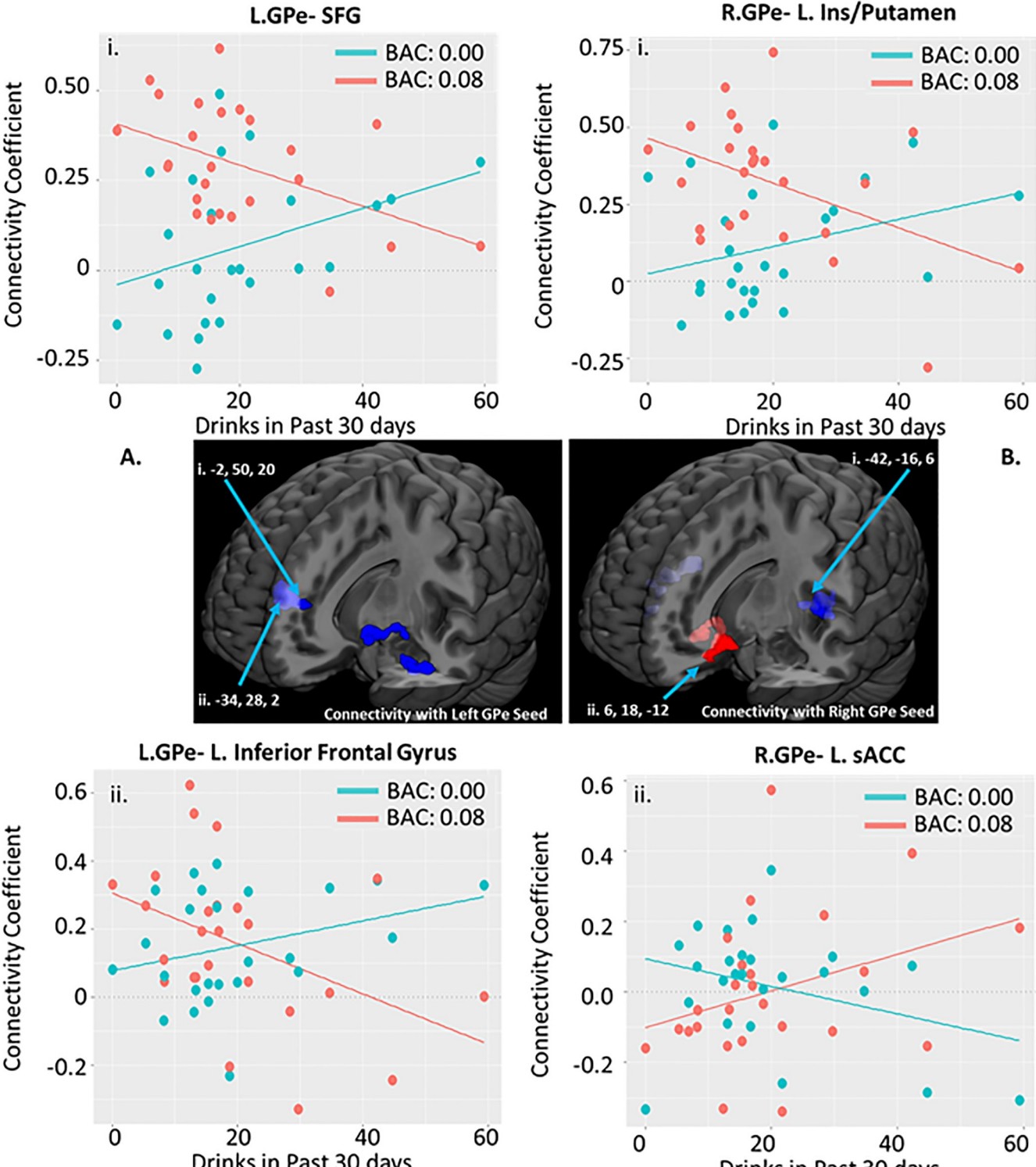

**Fig 2. Association between past 30 day drinking and alcohol induced changes in GPe connectivity.** (As measured by the Alcohol Timeline Followback). Scatter plots correspond to individual connectivity at baseline in teal (estimated BAC: 0.0) and after infusion in salmon (BAC: 0.08); linear fit lines are also displayed for each timepoint. Connectivity coefficient is the r (correlation value) between the signal in the GPe seed and the coordinate indicated. Brain images represent whole effects of drinks in past 30 days on alcohol induced change in functional connectivity (post-infusion connectivity > pre-infusion connectivity). Warm colors indicate intoxication related increases in connectivity; cool colors indicated intoxication related decreases in connectivity. Images shown at p < 0.005. Clusters with significant interaction effects are reported in Fig 3. Reported clusters of connectivity with GPe area correspond to the following regions: A.i.—Superior Frontal Gyrus extending into dorsal anterior cingulate; A.ii.—Hippocampus/amygdala; B.i.—Heschl's Gyrus extending into the insula/putamen; B.ii.—Heschl's Gyrus extending into the insula/putamen.

before each session. At the start of each visit, breath alcohol levels, urine pregnancy tests (if applicable) and urine drug screens were collected. In addition, we conducted a brief history interview about recent alcohol, medication, and nicotine use, changes in physical health, and menstruation (if applicable).

**Alcohol infusion to binge level exposure.** We performed the alcohol infusion procedure following previously published methods (fMRI and PET) [24, 25]. In both sessions, participants received an intravenous (IV) infusion of 6% v/v ethanol solution in saline to achieve a target breath alcohol concentration (BrAC) of 0.08 ± 0.005 g/dl at 15 min after the start of the infusion, and to maintain (or clamp) the target BrAC level for 30 min (Fig 1). The infusion-rate profile was computed for each subject using a physiologically-based pharmacokinetic (PBPK) model-based algorithm, with individualized estimates of the model parameters estimated from the subject's height, weight, age and sex [26, 27].

During the first session conducted outside the scanner, serial BrACs were measured at frequent intervals (5–15 minutes) using the Alcotest 7410+ handheld breathalyzer (Drager Safety Inc., CO), to ensure that each BrAC was within 0.005 g/dl of the target, and to enable minor adjustments to the infusion rates to overcome errors in parameter estimation and experimental variability [28, 29]. The adjusted infusion rate profile from the first session was used in the imaging session to replicate the target BrAC profile for each subject. This approach has been used to successfully achieve and maintain target BrACs, as verified by blood alcohol concentrations measured in samples drawn during the scan in other neuroimaging studies [25, 30]. After the end of the infusion, subjects were provided a meal, and BrAC was tracked until it dropped to 0.02 g/dL or below, at which time subjects were taken home by a designated driver or taxi.

**Neuroimaging acquisition and preprocessing.** During session 2, after the IV catheters were placed for alcohol infusion, participants were placed in a 3T SIEMENS Skyra MR scanner at the NMR Research Center at the NIH. Before starting the IV infusion of alcohol, a 5-min closed eyes resting-state fMRI scan was acquired with an echo-planar imaging sequence (36 axial slices, 3.8 mm thickness, 64 × 64 matrix and repetition time of 2000 ms, echo time 30, flip angle 90). After collecting this baseline resting-state fMRI scan, the alcohol infusion started and once the target BAC was achieved, a 10-minute waiting period was allowed to ensure the BAC was stable at the 0.08 g/dl level. During the waiting period, whole-brain structural and diffusion-weighted images of the brain were collected. Then, a second eyes-closed resting-state fMRI scan was collected while the target BAC (0.08 g/dl) was maintained. Following the MRI session, the Drug Effects Questionnaire was used to evaluate and quantify subjective effects of alcohol (DEQ) [31].

CONN ver.17f (http://www.conn-toolbox.org), a MATLAB/SPM-based (www.fil.ion.ucl.ac.uk/spm/) software, was used to conduct spatial/temporal preprocessing and analyses [32]. Spatial preprocessing included slice-timing correction, realignment, co-registration, normalization, and spatial smoothing (8 mm). Using the default settings in CONN, images were resliced into isotropic 2mm voxels. Anatomical volumes were segmented into grey matter, white matter, and cerebrospinal fluid (CSF) areas and the resulting masks were eroded to minimize partial volume effects. At the individual level, the temporal timeseries of the participant-specific six rotation-translation motion parameters and the timeseries from within the white matter/CSF masks were used as temporal covariates, and removed from the blood oxygen level-dependent (BOLD) functional data using linear regression. The resulting residual BOLD timeseries were band-pass filtered (0.008Hz < f < 0.15Hz) [33].

**Neuroimaging analysis.** To our knowledge there is no reported sub-division of human GPe with concrete landmark definition and/or Talairach coordinate demarcation. Therefore, for these analyses, we used right and left GPe masks obtained from the Lead-DBS

toolbox (http://www.lead-dbs.org/). These anatomical masks were generated using manual tracing of several structural MRIs and subsequent co-registration with histological studies, and have a combined volume of approximately 929 mm³ [34]. Seed-based correlations were calculated as Fisher-transformed bivariate correlation coefficients between the BOLD activity time-series in the seed ROI and each other individual voxel BOLD. CSF, grey matter, white matter signals and motion were included as covariates for this model at the single subject level. These correlations were calculated separately for the pre-infusion and post-infusion scans and for right and left GPe seeds. For each of the pre- and post-infusion analyses, 150 time points (across 300 seconds) were used.

To examine group level effects, the resulting single-subject connectivity maps were used as input for second level analyses in a general linear model. We modeled the within-subject effect of timepoint (i.e., post- versus pre-alcohol infusion; see S1 File, S1 Fig and S1 Table for report of the main effects of alcohol administration) and between-subject effects of impulsivity (BIS-11, UPPS-P, and DDT) and recent drinking behaviors (number of drinks in the last 30 days, number of heavy drinking days). S1 Table indicates clusters identified as showing significant changes from pre to post alcohol infusion, as well as associated statistics. For the seed-to-voxel analysis, reported clusters survived a height threshold of uncorrected $p < 0.001$ with a cluster level extent threshold of FDR-corrected $p < 0.05$. In Figs 3–5, these clusters are shown overlaid on structural brain cutaways. Corresponding line plots are only for visual interpretation of these effects. We also examined sex effects on alcohol-related changes in GPe connectivity at the whole brain level; there were no significant differences between males and females.

In order to test our hypothesis that the interaction between recent drinking behaviors and impulsivity would contribute to alcohol-induced changes in GPe functional connectivity, we extracted ROI-to-voxel correlation values between the seeds and significant clusters and used those values to conduct multiple regression analysis in R/RStudio (version 3.4.2/1.1.383). We

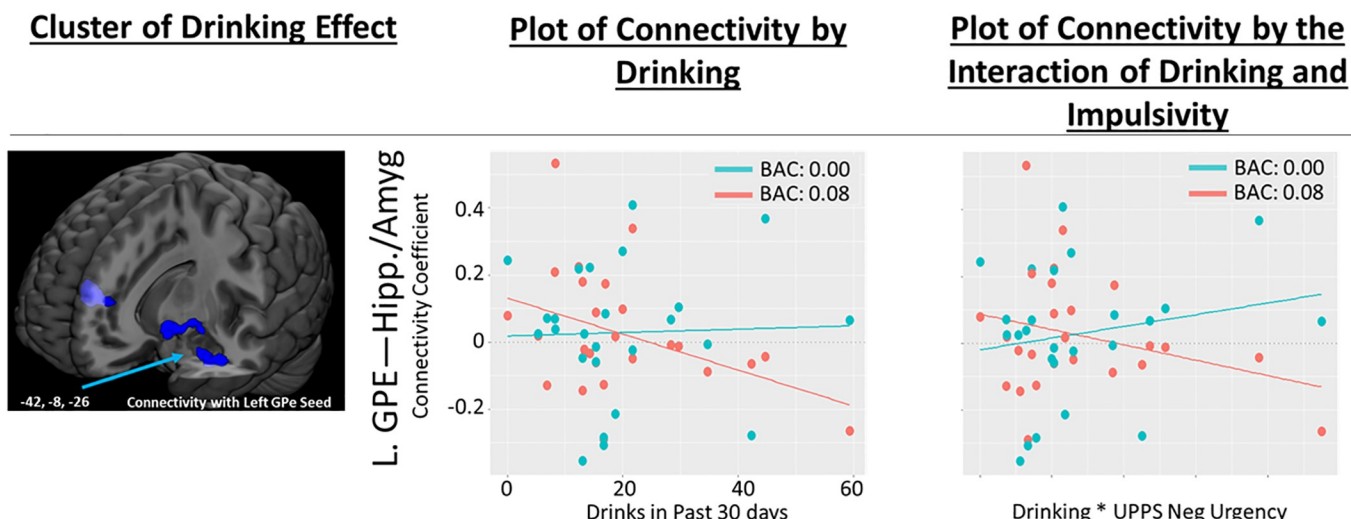

**Fig 3. Effect of alcohol induced changes in GPe connectivity on the association between drinking and impulsivity.** (As measured by the Alcohol Timeline Followback and UPPS-P, respectively). Cluster of Drinking Effect column: Brain images represent whole brain effects of drinks in past 30 days on alcohol induced change in functional connectivity (post-infusion connectivity > post-infusion connectivity). Warm colors indicate intoxication related increases in connectivity. Images shown at $p < 0.005$. Plots of Connectivity are scatter plots corresponding to individual connectivity at baseline in teal (estimated BAC: 0.0) and after infusion in salmon (BAC: 0.08); linear fit lines are also displayed for each timepoint. The y-axis for both plots is the same; the x-axis reflects the number of drinks in the last 30 days (center plot) and the number of drinks in the last 30 days multiplied by impulsivity score on the UPPS Negative Urgency scale (right plot). Abbreviations as follows: L. GPe- Left Globus Pallidus externus; Hipp./Amyg- Hippocampus/Amygdala cluster.

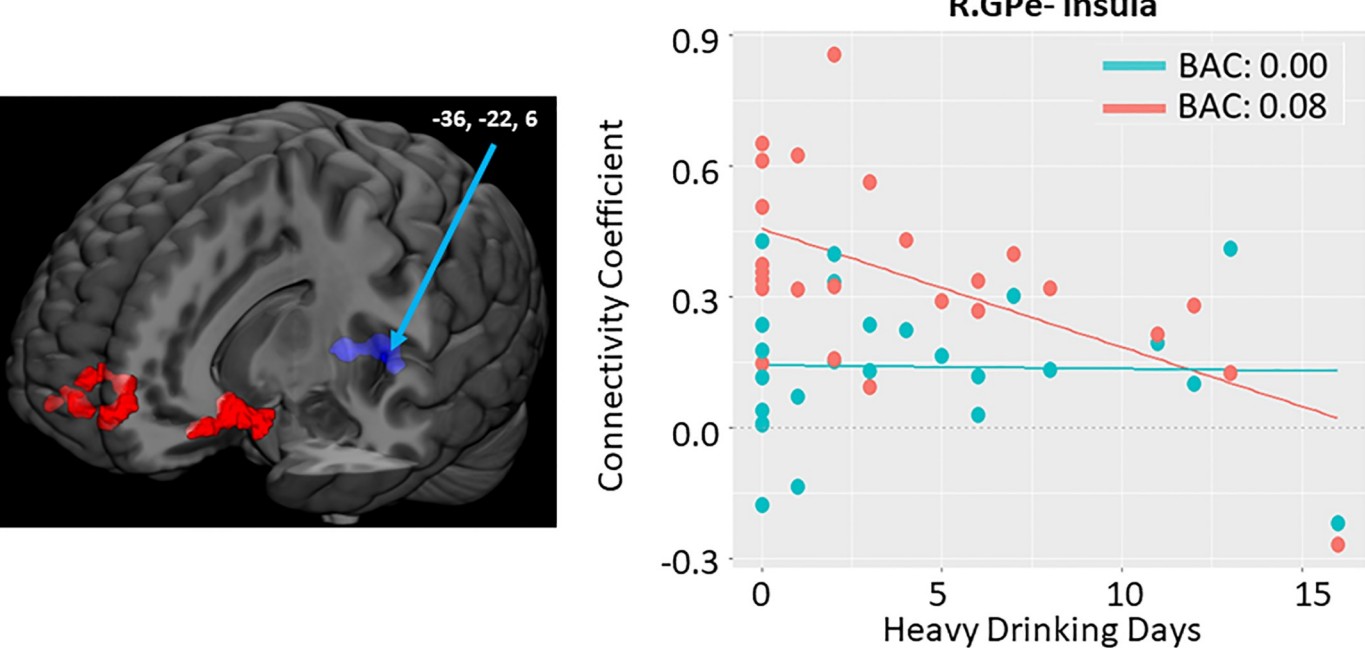

**Fig 4. Association between heavy drinking days and alcohol induced changes in GPe connectivity.** (As measured by the Alcohol Timeline Followback). Scatter plots correspond to individual connectivity at baseline in teal (estimated BAC: 0.0) and after infusion in salmon (BAC: 0.08); linear fit lines are also displayed for each timepoint. Connectivity coefficient is the r (correlation value) between the signal in the right GPe seed and the coordinate indicated. Brain images represent whole effects of drinks in past 30 days on alcohol induced change in functional connectivity (post-infusion connectivity > pre-infusion connectivity). Warm colors indicate intoxication related increases in connectivity; cool colors indicated intoxication related decreases in connectivity. Images shown at p < 0.005. Heavy Drinking Days as measured by the Alcohol Timeline Followback in the last 90 days. Abbreviations: R.GPe- Right Globus Pallidus Externus.

modeled drinking history, impulsivity, and the interaction between the two as independent variables on the dependent variable of change in GPe connectivity. We conducted these analyses only for significant clusters identified in the seed-to-voxel analysis, and only for the significant drinking and impulsivity measures. For the drinking measures identified in that seed-to-voxel analysis, models (for the specific significant cluster only) included the significant drinking variable as well as one of the two significant impulsivity variables; thus, 2 regressions per cluster were run. Similarly, for significant impulsivity measures, 2 regressions per cluster were run (as there are two drinking variables). This resulted in a total of 20 tests. We then adjusted the omnibus test statistics to correct for multiple comparisons using FDR of < 0.05. Of those regressions, only those with significant interaction effects are reported.

## Results

### GPe connectivity and drinking behaviors

Neither the total number of drinks over 30 days nor the number of heavy drinking days were correlated with subjective effects of the alcohol infusion as measured by the DEQ. Table 2 shows the correlations between drinking and impulsivity measures.

**Number of drinks in 30 days.** The number of drinks consumed per 30 days prior to screening was significantly associated with alcohol-related changes in connectivity (Fig 2 and Table 3). Drinks in past 30-days was negatively associated with alcohol-related changes in the connectivity between the left GPe and the left hippocampus /amygdala, left IFG extending into the anterior insula, and areas of the bilateral superior frontal / paracingulate cortices. Moreover,

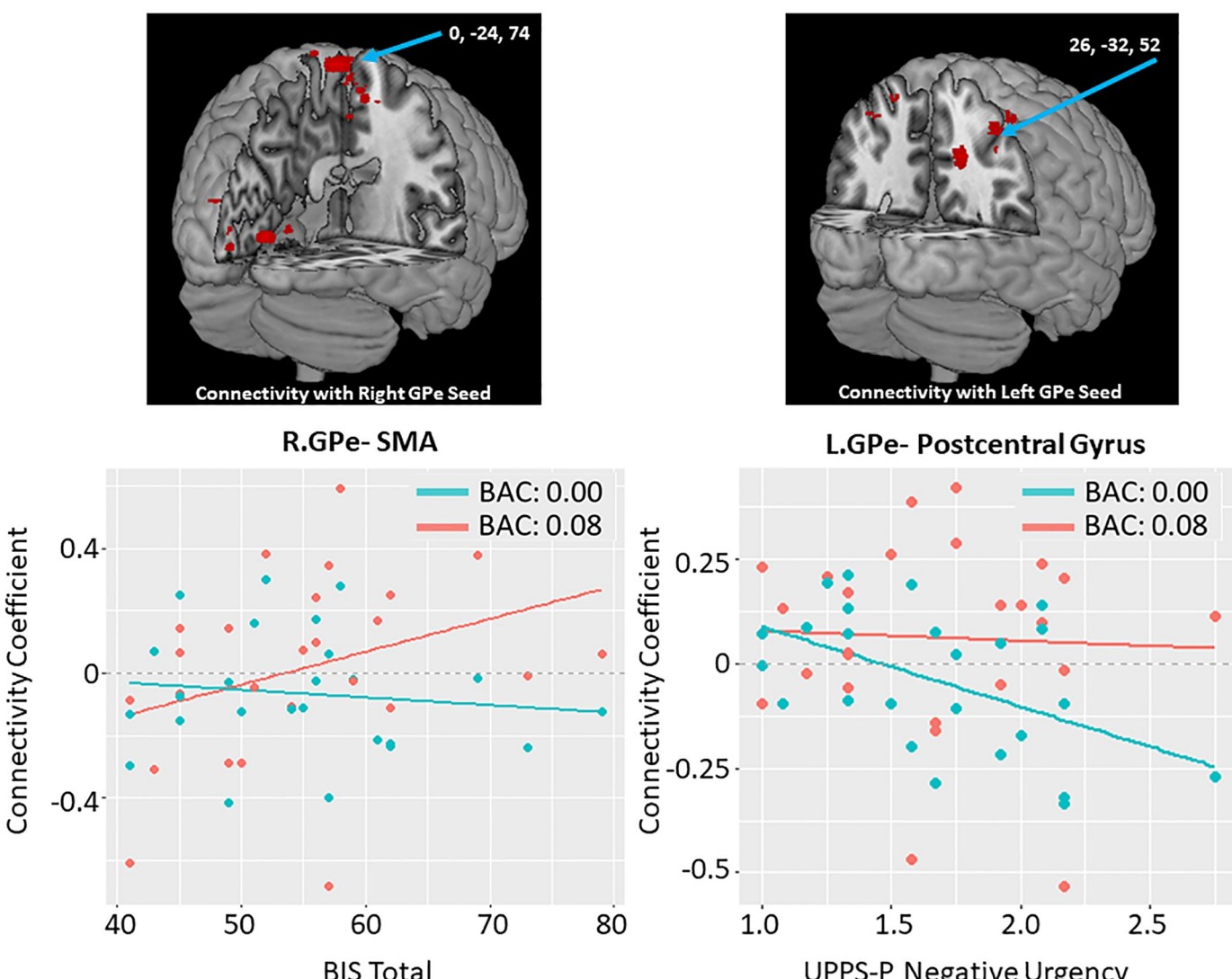

**Fig 5. Association between impulsivity measures and alcohol induced changes in GPe connectivity.** Scatter plots correspond to individual connectivity at baseline in teal (estimated BAC: 0.0) and after infusion in salmon (BAC: 0.08); linear fit lines are also displayed for each timepoint. Connectivity coefficient is the r (correlation value) between the signal in the GPe seed and the coordinate indicated. Brain images represent whole brain effects of drinks in past 30 days on alcohol induced change in functional connectivity (post-infusion connectivity > pre-infusion connectivity). Warm colors indicate intoxication related increases in connectivity. Images shown at p < 0.005. (left) Changes in connectivity plotted by total scores on the BIS (Barratt Impulsiveness Scale). (right) Changes in connectivity plotted scores on the UPPS-P negative urgency subscale. Reported clusters of connectivity with GPe area correspond to the following regions: (left) Precentral gyrus / SMA (supplementary motor area). Abbreviations: GPe- Globus Pallidus Externus; L.-Left; R.-Right.

the interaction between this drinking measure and negative urgency as measured by the UPPS-P significantly predicted left GPe-hippocampus/amygdala connectivity (F(3,21) = 4.694, omnibus FDR-p = 0.028, adjusted $R^2$ = 0.316, interaction $B$ = -0.023, interaction p = 0.008; Fig 3). Drinks in past 30-days was also negatively associated with alcohol-induced change in connectivity between right GPe and bilateral putamen, Heschl's gyrus, posterior insula, and central opercular cortices. For each of these associations, the change was such that in individuals with more past 30 days drinking, alcohol decreased connectivity; in individuals with less drinking, alcohol increased connectivity.

**Table 2. Correlations between drinking and impulsivity measures.**

| Measure | Delay Discounting Rate (K) | TLFB: Heavy Drinking Days | TLFB: Drinks Per Thirty Days | UPPS-P: Positive Urgency | UPPS-P: Negative Urgency | UPPS-P: Perseverance | BIS: Attentional Impulsiveness | BIS: Motor Impulsiveness | BIS: Nonplanning Impulsiveness |
|---|---|---|---|---|---|---|---|---|---|
| TLFB: Heavy Drinking Days | -0.22 | - | - | - | - | - | - | - | - |
| TLFB: Drinks Per Thirty Days | -0.33 | 0.79 *** | - | - | - | - | - | - | - |
| UPPS-P: Positive Urgency | 0.07 | 0.27 | 0.30 | - | - | - | - | - | - |
| UPPS-P: Negative Urgency | 0.22 | 0.30 | 0.14 | 0.48 * | - | - | - | - | - |
| UPPS-P: Perseverance | 0.06 | -0.28 | -0.22 | 0.06 | 0.46 * | - | - | - | - |
| BIS: Attentional Impulsiveness | 0.11 | 0.29 | 0.15 | 0.36 | 0.42 * | 0.13 | - | - | - |
| BIS: Motor Impulsiveness | 0.01 | 0.05 | 0.25 | 0.38 | 0.17 | 0.36 | 0.59 ** | - | - |
| BIS: Nonplanning Impulsiveness | 0.37 | -0.07 | -0.06 | 0.18 | 0.19 | 0.59 ** | 0.36 | 0.64 *** | - |
| BIS: Total | 0.22 | 0.09 | 0.11 | 0.36 | 0.30 | 0.46 * | 0.75 *** | 0.89 *** | 0.84 *** |

Abbreviations as follow: BIS- Barratt Impulsiveness Scale; TLFB- Alcohol Timeline Followback. Significance as follows-

*: $p < 0.05$;

**: $p < 0.01$;

***: $p < 0.001$

**Table 3. Associations between alcohol induced changes in GPe connectivity and drinking behaviors.**

| DRINKING BEHAVIOR CORRELATIONS | | Peak | | | Cluster Size | T | p-FDR | p-FWE | Anatomical Labels |
|---|---|---|---|---|---|---|---|---|---|
| | | x | y | z | | | | | |
| | Timeline Followback | | | | | | | | |
| | Right GPe x 30 day drinking (Negative correlation) | -42 | -16 | 6 | 100 | -6.04 | 0.023 | 0.014 | Left Heschl's Gyrus extending into the Insula, Central Operculum, Putamen |
| | | 42 | -20 | 10 | 83 | -4.96 | 0.029 | 0.036 | Right Heschl's Gyrus into the Insula, Planum Temporale, Central/Parietal Operculum, Putamen |
| | Right GPe x 30 day drinking (Positive correlation) | -12 | 18 | -10 | 98 | 6.57 | 0.008 | 0.016 | Left Subgenual Anterior Cingulate extending into the Accumbens |
| | | 6 | 18 | -12 | 75 | 6.56 | 0.015 | 0.057 | Right Subgenual Anterior Cingulate extending into the Accumbens |
| | Left GPe x 30 day drinking (Negative Correlation) | -2 | 50 | 20 | 142 | -6.01 | 0.001 | 0.001 | Superior Frontal Gyrus extending into dorsal anterior cingulate |
| | | -34 | 28 | 2 | 72 | -6.01 | 0.042 | 0.062 | Left Inferior Frontal Gyrus (pars triangularus) extending into Insula / Frontal Operculum |
| | | -42 | -8 | -26 | 66 | -6.14 | 0.042 | 0.089 | Left Hippocampus / Amygdala |
| | Right GPe x Heavy drinking days (Negative Correlation) | -36 | -22 | 6 | 147 | -6.08 | 0.001 | 0.001 | Left Insula extending into Heschls Gyrus, Putamen, Pallidum, and Planum Temporale |

**Table 4. Associations between alcohol induced changes in GPe connectivity and impulsivity measures.**

|  | Peak | | | Cluster Size | T | p-FDR | p-FWE | Anatomical Labels |
|---|---|---|---|---|---|---|---|---|
|  | x | y | z |  |  |  |  |  |
| *Left GPe x UPPS -P Negative Urgency* | 26 | -32 | 52 | 73 | 6.12 | 0.050 | 0.060 | Right Postcentral Gyrus |
| *Right GPe x BIS Total* | 0 | -24 | 74 | 96 | 5.73 | 0.021 | 0.019 | Precentral Gyrus (supplementary motor area) |

Clusters represent positive correlations between alcohol induced changes in connectivity and impulsivity measures; BIS: Barrett Impulsivity Scale; UPPS-P: Impulsivity Scale

The number of drinks consumed in the past 30 days was positively associated with alcohol-induced changes in connectivity between the right GPe and the bilateral subgenual cingulate extending into the basal ganglia. The change was such that in individuals with more past 30 days drinking, alcohol increased connectivity; in individuals with less drinking, alcohol decreased connectivity.

**Total number of heavy drinking days.** The total number of heavy drinking days was associated negatively with changes in right GPe connectivity with a left insula cluster extending into Heschl's gyrus, putamen, pallidum and planum temporale, such that there were no drinking pattern related differences in connectivity at baseline; however, in individuals with fewer heavy drinking days but not those with more, alcohol infusion increased connectivity (Fig 4 and Table 4).

## GPe connectivity and impulsivity

Measures of impulsivity (BIS-11, UPPS-P, DDT) were not correlated with subjective effects of alcohol infusion as measured by the DEQ. Although we did examine both positive and negative associations, there were no significant negative associations between alcohol-induced GPe connectivity changes and impulsivity. There were also no significant positive associations between alcohol-induced GPe connectivity changes and DDT. Other findings reported below and in Table 4 and Fig 5.

**BIS-11 trait impulsivity.** We found a significant positive association between BIS-11 total score and alcohol -induced connectivity change of right GPe with the precentral gyrus. In other words, higher impulsivity score was associated with increased GPe functional connectivity while under the influence of alcohol; there was no association between impulsivity and connectivity with that cluster at baseline. Specific BIS sub-scores were not significantly related to change in connectivity.

**UPPS-P trait impulsivity.** Tendency to display impulsive behaviors under strong negative emotions, as measured by the UPPS-P negative urgency score, was significantly positively associated with alcohol-induced connectivity changes between the left GPe and right postcentral gyri. There was little association between impulsivity and connectivity under the influence of alcohol, but more impulsivity was associated with less connectivity at baseline.

## Discussion

In this study, we examined the association between changes in GPe resting state connectivity following alcohol infusion with drinking history and impulsivity. Based on preclinical work finding alcohol alters neuronal firing in the GPe, and findings that the pathway is involved in response inhibition, we hypothesized that alcohol infusion would more greatly impact GPe functional connectivity in individuals with heavier drinking and more impulsivity. We also hypothesized that the interaction between recent drinking behaviors and impulsivity would

contribute to alcohol-induced changes in GPe functional connectivity. We found reductions in connectivity between the GPe and some subregions of the brain following alcohol infusion in individuals with more drinking in the past 30 days and more heavy drinking days. However, impulsivity was related to greater GPe connectivity while intoxicated. Heavier recent drinking behaviors in combination with higher impulsivity were associated with greater alcohol-induced connectivity change between GPe and amygdala regions, but for most of our connectivity findings, the contribution of drinking and impulsivity were distinct. Together, our findings suggest that the relationship between GPe-related neuronal firing and alcohol-related impulsivity is more complicated than we previously anticipated.

As predicted, we found that heavier drinking individuals had pronounced changes in GPe connectivity with a frontal gyrus regions. In these regions, individuals with patterns of heavy drinking had more coupling at baseline than individuals with lighter drinking patterns; in the binge infusion state, this association between heavy drinking history and GPe connectivity reversed. The alcohol-induced reduction in GPe-frontal pathways in heavier drinkers, as opposed to increased coupling, may specifically reflect impairment of stop signaling [35]. The inferior frontal gyrus (IFG) in particular has been demonstrated to control stop-signaling through the supplementary motor area, upstream from sub-cortical processes [36], possibly suggesting impaired behavioral inhibition or a shift towards reactive rather than proactive inhibition for individuals with more drinking [37]. This pattern may also reflect an alcohol sensitization effect in heavier drinkers, consistent with sensitization effects seen for other drugs of abuse, particularly in basal ganglia regions [38, 39]. It is also notable that increased connectivity between the frontal gryus and the GPe at rest has also been seen in individuals with hepatic encephalopathy, a result of liver disease such as alcoholic cirrhosis [40]. Our new findings may suggest these previously observed baseline differences reflect a pre-disease vulnerability in heavier drinkers, or indeed that the effects of alcohol even at a non-problem level on the liver contributes to damage to this pathway.

We also found that heavier drinking individuals had less change caused by alcohol infusion in connectivity between the GPe and dorsal ACC, superior frontal gyrus, and insula regions. In fact, while alcohol decreased GPe connectivity with these regions in individuals with more recent drinking, the effect was reverse for those with less drinking. These regions are thought to play a role in signaling unsuccessful stopping, particularly in cases where errors are less common [41, 42]. Intoxication-related changes in connectivity between this region and the GPe in individuals with less recent drinking may reflect novelty of failures in stop-signaling. Individuals who drink more may expect failures of stop-signaling during intoxication, as opposed to those with infrequent drinking. A lack of baseline association between recent drinks and error feedback in the right insula is consistent with lack of differences in expectation of low-frequency errors at baseline. Alternatively, during intoxication, less stop-signaling could correspond to less failures, leading to a reduced need for error signaling. However, there may be other explanations; the insula in particular is a heterogeneous structure involved in a variety of functions. For example, Feng and colleagues suggest reductions in spontaneous activity in this posterior insula structure reflect input of emotional state feedback during fear memory consolidation [43]. From that perspective, one might interpret our results to mean that in individuals with less recent drinking, alcohol decreases the integration of emotional state feedback into behavior.

On the other hand, we found that alcohol increased subgenual ACC-GPe connectivity in individuals with more recent drinking. This pathway may be involved in reward-driven overrides of the stop-signal [44]. Greater coupling of ACC-GPe at baseline may lead to a greater tendency to override the stop-signal in face of drinking; this would likely be more pronounced in individuals with positive experiences with alcohol, such as light social drinking, or from the

rewarding nature of novelty. However, this override would be less necessary when the impulse control pathways are working sub-optimally and sending less stop-signals, such as during intoxication. Alcohol intoxication may also serve to reduce inhibition via increased connectivity to this override; our results may suggest alcohol has a reward heightening effect in individuals with more recent drinking but a dampening effect in lighter drinkers.

This fronto-striatal circuit has also been described as subserving development of cognitive and motor behavioral plans [45]; an alternative explanation for our findings, then, given baseline decoupling in individuals with more recent drinking, could be a general deficit in action planning. However, it is difficult to determine if lower baseline GPe connectivity is caused by heavy drinking or if it is a vulnerability factor that leads to heavy drinking. It may be that individuals with lower GPe-ACC connectivity are more vulnerable to engaging in heavy drinking episodes. Another possibility is that drinking, even at a social level, has lasting effects on resting thresholds of GPe connectivity; for example, some work has shown that alcohol leads to hyperactive IFG response during stop-signaling [18]. Future work should use longitudinal approaches to investigate risk factors and consequences of heavy and binge drinking across the lifespan.

We also found that heavier drinking individuals had greater decoupling caused by alcohol infusion in connectivity between the GPe and amygdala/hippocampus regions. The amygdala modulates striatum activity during reward learning and instrumental choice through direct connections within the limbic stream [35]. Moreover, connectivity between the amygdala and basal ganglia is associated with greater stress-induced cortisol response dependent on the availability of mineralocorticoid receptors [46]. Decoupling of the amygdala and GPe signaling during intoxication in heavier drinkers, particularly those with a tendency to act impulsively during negative emotions, may reflect drinking to reduce stress rather than drinking as a reward. This "drinking to reduce negative affect" is a key feature in addiction [47], and may mean that individuals with this pattern of response are particularly vulnerable to developing alcohol use disorder. In fact, chronic alcohol consumption leads to increased mineralocorticoid receptor pathway activity [48], suggesting that by drinking to reduce stress, these individuals may be making themselves more vulnerable to stress in the future.

Impulsivity across measures was associated with greater change in GPe connectivity following alcohol infusion. For clusters in the postcentral and precentral gyrus, individuals low on impulsivity had no differences in connectivity with the GPe as a result of alcohol. For the precentral/supplementary motor area(SMA) cluster, across the continuum of impulsive traits, there were no baseline differences in GPe connectivity, but after alcohol infusion, impulsivity was related to increased connectivity of these areas to the GPe. Previous work has shown that increased SMA-dorsal striatum structural connectivity was associated with more motor impulsivity, such that these results may suggest that individuals with trait impulsivity have an alcohol-related vulnerability to greater behavioral impulsivity [49]. Alternatively, SMA engagement is associated with reactive inhibition [37] and is thought to influence inhibition through a hyperdirect rather than indirect pathway [50]. However, the highest instance of successful inhibition is when the hyperdirect and indirect pathways work in conjunction [12], meaning impulsive individuals with disproportional recruitment of the hyperdirect pathway, as seen here may still have impaired inhibition.

On the other hand, in the postcentral cluster, negative urgency was associated with hypo-connectivity to the GPe at baseline, but not when intoxicated. The postcentral gyrus is part of a motor cortical network activated during "go" signalling; it may be that individuals higher in impulsivity have a lower threshold for such a signal, reflected in the lower baseline value [51]. If so, intoxication may compound the risk for impulsive behavior in these individuals. An alternative explanation is that individuals with higher negative urgency tend to engage in internal rumination more; the previous literature illustrates that decreases in activation in the

postcentral gyrus at rest are more pronounced in interospective, eyes-closed conditions. Alcohol may then serve to normalize this self-focused dwelling on internal feelings in individuals higher in negative urgency [52].

## Limitations

Because we did not have a placebo infusion condition, we cannot rule out that pre- to post-infusion changes were not due to dynamic fluctuations within resting states. However, we do not believe this is the case. A previous study using an alcohol infusion paradigm with a full, double-blind crossover design found alcohol effects on resting-state connectivity compared to placebo [53]. Although that study examined network level connectivity changes rather than seed-based changes in subcortical regions as reported here, the fact that acute alcohol intoxication impacts connectivity during resting state has already been established. Moreover, this study is based on animal literature that found alcohol-induced changes in this pathway that are dose-dependent [17].

Our measures of impulsivity and drinking were primarily retrospective and obtained via self-report. Although we have designed our analysis to provide insight on the relationship between impulsivity, drinking, and alcohol-related changes in brain connectivity, we cannot comment on the directionality or causality of these relationships. Nonetheless, we did find evidence that connectivity during our laboratory "binge drinking" state was influenced by previous drinking and impulsive traits. Future studies should take a prospective approach that could speak directly to the neural mechanisms leading to problem drinking behaviors, as well as measuring within-subject changes in choice impulsivity during acute intoxication.

Another limitation to our inferences about baseline impulsivity and drinking measures is that our participants were light social drinkers with a limited range of alcohol use and an absence of AUD. This may explain why impulsivity and drinking measures were not significantly correlated in our sample, despite their clear link in the literature (e.g., [54]). Overall, our findings are most applicable to non-AUD binge drinking groups, such as adolescents and emerging adults, rather than in chronic heavy binge drinkers with AUD.

The small sample size of this study (n = 25) allowed us to test our specific hypotheses but may limit our ability to rule out the involvement of GPe connectivity to areas that were not found to be significant in this study. We limited our investigation to the connectivity of the hypothesized region of interest (GPe) and did not investigate other potential pathways of behavior control. Importantly, we do not assert that GPe pathways are the only mechanisms of control influenced by acute or chronic alcohol consumption or related to impulsivity. In fact, increased connectivity in behavioral control circuits suggests that alternative mechanisms or alternative recruitment of pathways are involved in failures of inhibition. Although outside the scope of the current study, future work should investigate those pathways during acute alcohol intoxication in a larger sample, and during specific stop-signaling tasks.

Because we do not have behavioral data on response inhibition in the alcohol intoxication state, there are other potential explanations for our findings. First, given that our study looked at resting-state connectivity differences, rather than actual differences in stop-signaling, it is possible that observed changes are not directly related to behavioral control. Abnormal frontostriatal connectivity has also been implicated in anhedonia and psychomotor retardation [55], mental time-keeping [56], cognitive load during working memory [57], set-shifting [58], and long-term reward learning [59]. Changes in connectivity could also be driven by alcohol-related changes in neurotransmitter availability across the whole brain that may or may not translate directly to functional change; dopamine has been shown to modulate both resting and set-shifting based functional connectivity [60].

## Conclusion

Here we translated preclinical research to investigate how alcohol-induced GPe connectivity disruptions explain impulsive behaviors and drinking patterns in social drinkers. We found that impulsivity and alcohol use are associated at rest with impairment of GPe circuits implicated in typical stop-signaling processing. To our knowledge, this is the first demonstration of an association between impulsivity, drinking measures, and GPe connectivity changes under clamped alcohol exposure in humans. These findings suggest that individuals with impulsive traits and past drinking are particularly vulnerable to the control impairments of alcohol intoxication. Moreover, they have important implications with regard to identifying those most at risk for drinking-related consequences; they suggest a mechanism for how alcohol leads to impulsive behaviors, including excessive drinking. Finally, the neuronal connectivity changes induced by alcohol may represent a target for interventions that would mitigate the impact of binge drinking or AUD by reducing impulsivity. Although additional work, including longitudinal studies, are needed before these findings can be useful in a clinical setting, this study achieves the important first step of translating preclinical models of binge drinking to the human brain and behavior.

## Supporting information

**S1 File. Alcohol effects on GPe connectivity.** Results and discussion of the main effects of alcohol on GPe connectivity.
(PDF)

**S1 Fig. Results of the main effects analysis of post versus pre alcohol infusion connectivity.**
*(A)* Shows the first 50 time points of the timeseries of the preprocessed BOLD signal from the right GPe (averaged across voxels within the right GPe and across subjects; sampled every two seconds), before (pre) and after (post) the IV-alcohol infusion. *(B)* Change in the GPe whole brain connectivity (Post vs. Pre contrast). Regions that decreased connectivity with the GPe after IV-alcohol infusion are in blue; regions with increased connectivity are in red; GPe masks are in green.
(TIF)

**S1 Table. Alcohol induced changes in GPe connectivity.**
(XLSX)

## Author Contributions

**Conceptualization:** Karina P. Abrahao, Carlos R. Cortes, David M. Lovinger, Reza Momenan.

**Data curation:** Melanie L. Schwandt, Reza Momenan.

**Formal analysis:** Samantha J. Fede, Carlos R. Cortes.

**Funding acquisition:** Reza Momenan.

**Investigation:** Carlos R. Cortes, Reza Momenan.

**Methodology:** Vijay A. Ramchandani.

**Project administration:** Reza Momenan.

**Resources:** David T. George, Nancy Diazgranados, Reza Momenan.

**Supervision:** David M. Lovinger, Reza Momenan.

**Visualization:** Samantha J. Fede.

**Writing – original draft:** Samantha J. Fede, Carlos R. Cortes.

**Writing – review & editing:** Samantha J. Fede, Karina P. Abrahao, Carlos R. Cortes, Erica N. Grodin, Melanie L. Schwandt, David T. George, Nancy Diazgranados, Vijay A. Ramchandani, David M. Lovinger, Reza Momenan.

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
