## [Decision Letter · Decision Letter 0]

10 Jan 2020

PONE-D-19-29169

Alcohol Effects on Globus Pallidus Connectivity: Role of Impulsivity and Binge Drinking

PLOS ONE

Dear Dr. Momenan,

Thank you for submitting your manuscript to PLOS ONE. After careful consideration, we feel that it has merit but does not fully meet PLOS ONE’s publication criteria as it currently stands. Therefore, we invite you to submit a revised version of the manuscript that addresses all the points raised during the review process.

In particular, regarding the statistical analyses and presentation of the data: 1) present effect sizes in the results section and include analyses on gender effects as suggested by reviewer 1 and 2) use more conservative cluster-forming thresholds and be more specific and complete in the description of the employed analyses as suggested by reviewer 2.

In addition, the GPe as region of interest should be more clearly motivated as suggested by reviewer 1, and the discrepancy between the discussion of behavioural data and the absence thereof in the results section should be addressed.

We would appreciate receiving your revised manuscript by Feb 24 2020 11:59PM. To enhance the reproducibility of your results, we recommend that if applicable you deposit your laboratory protocols in protocols.io, where a protocol can be assigned its own identifier (DOI) such that it can be cited independently in the future. For instructions see: http://journals.plos.org/plosone/s/submission-guidelines#loc-laboratory-protocols

We look forward to receiving your revised manuscript.

Kind regards,

Tommy Pattij, Ph.D.

Academic Editor

PLOS ONE

Journal Requirements:

2. Our internal editors have looked over your manuscript and determined that it may be within the scope of our Neuroscience of Reward and Decision Making Call for Papers. This collection of papers is headed by a team of Guest Editors for PLOS ONE: Stephanie Groman, Satoshi Ikemoto, Jane Taylor and Robert Whelan. With this Collection we hope to bring together researchers working on a wide range of disciplines, from animal subjects research, computational approaches and patient-centered research. Additional information can be found on our announcement page: https://collections.plos.org/s/reward-and-decision-making. If you would like your manuscript to be considered for this collection, please let us know in your cover letter and we will ensure that your paper is treated as if you were responding to this call. Agreeing to be part of the call-for-papers will not affect the date your manuscript is published. If you would prefer to remove your manuscript from collection consideration, please specify this in the cover letter.

Additional Editor Comments (if provided):

Reviewers' comments:

Reviewer's Responses to Questions

**Comments to the Author**

1. Is the manuscript technically sound, and do the data support the conclusions?

Reviewer #1: Yes

Reviewer #2: Yes

2. Has the statistical analysis been performed appropriately and rigorously? 

Reviewer #1: Yes

Reviewer #2: I Don't Know

3. Have the authors made all data underlying the findings in their manuscript fully available?

Reviewer #1: Yes

Reviewer #2: No

4. Is the manuscript presented in an intelligible fashion and written in standard English?

Reviewer #1: Yes

Reviewer #2: Yes

5. Review Comments to the Author

Reviewer #1: This is an interesting and relevant study that focus on the harm caused by binge drinking in young men and women. To better understand the neural mechanisms leading to risky and disinhibited intoxication-related behaviors the authors tested twenty-five healthy volunteers who underwent intravenous alcohol infusion to achieve a blood alcohol level of 0.08 g/dl. A resting state functional magnetic resonance imaging scan was collected prior to the infusion and at binge-level exposure. The authors were in particular interested in assessing alcohol-induced changes in globus pallidus externus (GPe) connectivity in relation to drinking behaviors and impulsivity traits. The authors found that subjects with greater number of drinks in the recent past had greater alcohol-induced deficits in the connectivity of GPe particularly to the striatum; impulsivity was related to greater GPe connectivity while intoxicated.

Overall this is a well written paper. However, I have some concerns, which are listed below:

One of the main problems of this study is the relatively small sample size for the number of analyses performed. The authors may provide effect sizes for their significant results. Sex effects were not explored although half of the participants were women. The menstrual cycle can have an influence on the metabolization of alcohol and although it was recorded in the study, it is not noted if it was controlled for or considered in the data analyses. The targeted blood alcohol level of 0.08 g/dl may have different effects on neurofunctional connectivity in men and women depending on individual alcohol tolerance levels. Nicotine consumption should also be used as a co-variate. The methods are well described; however, it is not clear why the authors focused on the globus pallidus externus (GPe) and did not use other seed regions involved in inhibitory control as well. No behavioral performance data are presented, yet the authors extensively discuss the stop-signal task along the lines of “reduced behavioral inhibition” or “reactive rather than proactive inhibition for individuals with more drinking.” Either the discussion needs to be reformulated or performance data should be added to the analyses. The list of limitations of the study is long including not having a placebo group and a small sample size. However, because of the complexity of the results, the introduction and discussion would profit from a model that can integrate the different connectivity results and also provide a rationale for focusing on the GPe as seed region for the analyses.

Reviewer #2: “Alcohol effects on globus pallidus connectivity: role of impulsivity and binge drinking” by Dr. Fede and colleagues describes (clamped) alcohol-induced alterations in resting-state connectivity of the globus pallidus externus (GPe) with other regions, and how that relates to recent drinking history and impulsivity traits/behavior. This well-written manuscript is interesting and timely. The most major critique relates to the cluster-forming threshold chosen. A few points could be clarified with greater detail, elaborated below.

Major

Line 215: A cluster-forming threshold of p<.005, pFDR<.05 is rather liberal (Eklund et al. 2016), with cluster-forming thresholds of p<.001 producing much more acceptable false-positive rates. Unless there is a rationale for using a more liberal threshold with these connectivity data, the p<.001 threshold should be followed.

Lines 220-231: If I follow, the authors performed paired t-tests (pre, post) with no regressors, then paired-t tests with impulsivity and drinking regressors. These clusters’ beta weights were then extracted and used as the outcome of a linear model conducted in R(?) Please clarify. Be very explicit about exactly how the analyses were performed, with a helpful step-by-step description in more depth. This section was a bit hard to follow.

Line 343: “As predicted, we found that heavier drinking individuals had more pronounced changes in GPe connectivity with the basal ganglia, OFC, and mPFC regions. In these regions, individuals with patterns of heavy drinking had less coupling at baseline than individuals with lighter drinking patterns;” Figure 2 (upper right) appears to illustrate the heaviest drinkers with more coupling at baseline (r= ~.25) relative to the lightest drinkers (r= ~0) in the R.GPe-L insula/putamen, with the lighter drinkers showing the largest alcohol-induced change in connectivity. Please clarify

Minor

Line 56: “This is largely due to organ damage caused by chronic drinking, intoxication-related car accidents and domestic violence, and the development of alcohol use disorder (AUD) [3, 4].” AUD is antecedent to the vast majority of the ills described here; please clarify this relationship.

Table 1: Though it is specified in-text, it would be helpful to discriminate the two TLFB measures; either in the table Notes or within the table, e.g., “Drinks in last Thirty Days”.

Line 133: What monetary amounts were used for the DDT?

Supplementary Figure 1A: Perhaps I misunderstand, but the way this is presented it appears to show a (group mean?) time series from the right GPe OFF and ON alcohol. If so, then the stereotyped pattern from the (pre) condition resting scan is replicated in a (post) resting scan 25 min later, albeit with a small timeshift. It seems impossible that a resting scan would produce a stereotyped/replicated pattern as illustrated; i.e. I would expect that the group mean resting BOLD signal would look like noise. Please explain what this figure is showing.

Interestingly, alcohol-increased connectivity was mostly ipsilateral for left GPe, but mostly contralateral for right GPe. What do the authors make of this?

Line 239: “The number of drinks consumed per 30 days prior to screening had a significant effect on alcohol-related change in connectivity.” Please reword to reflect “association”; the current wording implies directionality and causation.

Figure 2 legend: “B.i. Heschl’s Gyrus…” appears to be labeled “ii.” on the figure diagram.

Figure 3 legend: “Reported clusters clusters of connectivity with GPe area correspond to the following regions: i. – Orbitofrontal cortex extending into subgenual anterior cingulate; ii. – Angular/supramarginal gyrus; iii. Frontal pole.” It seems that “GPe area” should be described as “left GPe”. No mention is made of “iv”, “v”, or “vi”.

Figure 5: “Diagram of Drinking/Impulsivity Mediation column” is not present in the attached figures. Only 3 columns are displayed in Fig 5.

6. PLOS authors have the option to publish the peer review history of their article (what does this mean?). If published, this will include your full peer review and any attached files.

Reviewer #1: No

Reviewer #2: No

---

## [Author Response · Author response to Decision Letter 0]

4 Feb 2020

Reviewer #1: 

1) One of the main problems of this study is the relatively small sample size for the number of analyses performed. The authors may provide effect sizes for their significant results.

Response: We have added T-values to Tables 3 & 4 and report adjusted R2 for the regression analysis.

2) Sex effects were not explored although half of the participants were women. The menstrual cycle can have an influence on the metabolization of alcohol and although it was recorded in the study, it is not noted if it was controlled for or considered in the data analyses. The targeted blood alcohol level of 0.08 g/dl may have different effects on neurofunctional connectivity in men and women depending on individual alcohol tolerance levels.

Response: We ran a supplementary analysis to address this concern and noted the lack of sex effects in the manuscript as follows (page 10): “We also examined sex effects on alcohol-related changes in GPe connectivity at the whole brain level; there were no significant differences between males and females.”

3) Nicotine consumption should also be used as a co-variate. 

Response: There was only one participant in our study who reported smoking (as indicated in Table 1). Therefore, we cannot covary nicotine consumption, but we do not expect it influences our results.

4) The methods are well described; however, it is not clear why the authors focused on the globus pallidus externus (GPe) and did not use other seed regions involved in inhibitory control as well. 

Response: We focus on the GPe because our study was designed to translate the preclinical findings of Abrahao et al., 2017, which found that acute alcohol administration increased the firing rate of neurons in the GPe that project efferently. Based on this, we had reason to hypothesize that alcohol administration would alter GPe functional connectivity. This is described in the introduction (pages 3-4) and further, we have added the following language to emphasize this: “In this study, we aimed to extend the preclinical results of the Abrahao et al, 2017 study to a human paradigm by…”

5) No behavioral performance data are presented, yet the authors extensively discuss the stop-signal task along the lines of “reduced behavioral inhibition” or “reactive rather than proactive inhibition for individuals with more drinking.” Either the discussion needs to be reformulated or performance data should be added to the analyses. 

Response: As explained in our response to comment #4, the aims of our analyses were to test in a human model alcohol-related changes in GPe pathways. These pathways are theorized to be involved in response inhibition/stop-signaling (e.g., Jahfari et al., 2011). Moreover, based on previous literature (i.e., Nikolaou et al., 2013) we know that alcohol impairs GPe function during stop-signaling. Although we do not have data on changes in stop-signal neural response during intoxication (which we acknowledge in the limitations on page 20), a large portion of our analysis focused on behavioral impulsivity and trait impulsivity measures. We, therefore feel the stop-signaling/response inhibition oriented discussion of our results is the best explanation of our findings in the context of the prior literature. We do, however, acknowledge that there may be other ways to interpret our findings (page 20).

6) The list of limitations of the study is long including not having a placebo group and a small sample size. However, because of the complexity of the results, the introduction and discussion would profit from a model that can integrate the different connectivity results and also provide a rationale for focusing on the GPe as seed region for the analyses.

Response: In concurrence with this Reviewer’s other concerns, we have emphasized our motivation to translate the results of a mouse-model published in Abrahao et al. 2017 vis the framework of approaching alcohol-related GPe signaling in the context of the indirect pathway of response inhibition. We believe this framework effectively integrates the results. We have added the following language to the first paragraph of the discussion to further emphasize this as follows (page 15): “Based on preclinical work finding alcohol alters neuronal firing in the GPe, and findings that the pathway is involved in response inhibition, we hypothesized that…”

Reviewer #2: 

Major

1) Line 215: A cluster-forming threshold of p<5, pFDR<.05 is rather liberal (Eklund et al. 2016), with cluster-forming thresholds of p<.001 producing much more acceptable false-positive rates. Unless there is a rationale for using a more liberal threshold with these connectivity data, the p<.001 threshold should be followed.

Response: We have updated our analysis to use the p<.001 cluster forming threshold per the reviewer’s recommendation. The results, discussion, table, figures, and supplementary materials have been updated accordingly. 

2) Lines 220-231: If I follow, the authors performed paired t-tests (pre, post) with no regressors, then paired-t tests with impulsivity and drinking regressors. These clusters’ beta weights were then extracted and used as the outcome of a linear model conducted in R(?) Please clarify. Be very explicit about exactly how the analyses were performed, with a helpful step-by-step description in more depth. This section was a bit hard to follow.

Response: We have updated our methods to more clearly describe the Conn first and second-level processing pipeline, and our process in extracting the correlation values for use investigating the interaction between drinking history and impulsivity in the regression model in R. (pages 9/10)

3) Line 343: “As predicted, we found that heavier drinking individuals had more pronounced changes in GPe connectivity with the basal ganglia, OFC, and mPFC regions. In these regions, individuals with patterns of heavy drinking had less coupling at baseline than individuals with lighter drinking patterns;” Figure 2 (upper right) appears to illustrate the heaviest drinkers with more coupling at baseline (r= ~.25) relative to the lightest drinkers (r= ~0) in the R.GPe-L insula/putamen, with the lighter drinkers showing the largest alcohol-induced change in connectivity. Please clarify

Response: We thank the reviewer for pointing out this inconsistency. We have revised the discussion somewhat based on our updated results, and have paid particular attention to making sure this paragraph (now page 16) more accurately reflects the findings.

Minor

4) Line 56: “This is largely due to organ damage caused by chronic drinking, intoxication-related car accidents and domestic violence, and the development of alcohol use disorder (AUD) [3, 4].” AUD is antecedent to the vast majority of the ills described here; please clarify this relationship.

Response: We have rephrased the sentence as follows: “This is largely due to the development of alcohol use disorder (AUD) and consequential organ damage caused by chronic drinking, intoxication-related car accidents and domestic violence [3, 4].”

5) Table 1: Though it is specified in-text, it would be helpful to discriminate the two TLFB measures; either in the table Notes or within the table, e.g., “Drinks in last Thirty Days”.

Response: We have added the following to the Table 1 Note: “Heavy drinking days was calculated from the last 90 days as measured by the TLFB, where a heavy drinking day was defined as 4 or more drinks per day for women and 5 or more drinks per day for men. Drinks per thirty days was the total count of drinks from the past 30 days as measured by the TLFB.”

6) Line 133: What monetary amounts were used for the DDT? 

Response: We have added the following to the description of the DDT (page 6): “Subjects were presented as their immediate reward values between $100 and $0 dollars in increments of $10, while the delayed reward was $100. Delays were 0, 7, 14, 20, 25, or 30 days. Subjects completed trials with all iterations of these parameters.”

7) Supplementary Figure 1A: Perhaps I misunderstand, but the way this is presented it appears to show a (group mean?) time series from the right GPe OFF and ON alcohol. If so, then the stereotyped pattern from the (pre) condition resting scan is replicated in a (post) resting scan 25 min later, albeit with a small timeshift. It seems impossible that a resting scan would produce a stereotyped/replicated pattern as illustrated; i.e. I would expect that the group mean resting BOLD signal would look like noise. Please explain what this figure is showing.

Response: This is in fact the time series of the first 50 time points of the resting state scan in the ON and OFF alcohol conditions This time series is averaged across voxels in the seed region of interest (right GPe) and across subjects. This time series is also post-preprocessing, so noise reduction steps have already been carried out as reflected in this time course. We have added the following description to the caption of the supplementary figure as follows: “Shows the first 50 time points of the timeseries of the preprocessed BOLD signal from the right GPe (averaged across voxels within the right GPe and across subjects…”

8) Interestingly, alcohol-increased connectivity was mostly ipsilateral for left GPe, but mostly contralateral for right GPe. What do the authors make of this?

Response: We have added this discussion to the supplementary materials: “Alcohol has previously been shown to reduce the lateralization of specific functions, particularly in terms of greater left lateral increases and right lateral decreases in connectivity at rest (Volkow et al., 2008). The greater left-side increases in connectivity (both from ipsi- and contralateral GPe regions) may reflect this pattern as well. We would point out that there were mostly contralateral alcohol-decreases in connectivity from the left GPe, suggesting reductions in connectivity on the right side. We would also point out from the perspective of the role of the GPe pathways, previous work finds that connectivity strength in right-lateralized hyperdirect and indirect basal ganglia/frontal pathways predicted successful response inhibition (Jahfari et al., 2011). On the other hand, increases in left ipsilateral connectivity in individuals with chronic alcohol use has been associated with compensatory function (Chanraud et al., 2011). Taken together, the lateralization pattern may reflect alcohol related impairment of standard inhibitory pathways and increases in “alternative” neural communication routes.” 

9) Line 239: “The number of drinks consumed per 30 days prior to screening had a significant effect on alcohol-related change in connectivity.” Please reword to reflect “association”; the current wording implies directionality and causation.

Response: We have reworded this to: “The number of drinks consumed per 30 days prior to screening was significantly associated with alcohol-related changes in connectivity.”

10) Figure 2 legend: “B.i. Heschl’s Gyrus…” appears to be labeled “ii.” on the figure diagram.

Response: We have corrected this to be labeled “i.” on the diagram.

11) Figure 3 legend: “Reported clusters clusters of connectivity with GPe area correspond to the following regions: i. – Orbitofrontal cortex extending into subgenual anterior cingulate; ii. – Angular/supramarginal gyrus; iii. Frontal pole.” It seems that “GPe area” should be described as “left GPe”. No mention is made of “iv”, “v”, or “vi”.

Response: We have updated the figure legend to correspond to the updated results based on requested revisions; as such, Figure 3 is now Figure 4. Figure 4 and its legend no longer refer to roman numerals, and we have specified the laterality of the GPe seed.

12) Figure 5: “Diagram of Drinking/Impulsivity Mediation column” is not present in the attached figures. Only 3 columns are displayed in Fig 5.

Response: We have corrected this error by updating the note to reflect the current version of the figure. Note that the order of the figures has been updated based on the requested revisions and Fig 5. Is now Fig. 3.

---

## [Decision Letter · Decision Letter 1]

19 Feb 2020

PONE-D-19-29169R1

Alcohol Effects on Globus Pallidus Connectivity: Role of Impulsivity and Binge Drinking

PLOS ONE

Dear Dr. Momenan,

Thank you for submitting your manuscript to PLOS ONE. After careful consideration, we feel that it has merit but does not fully meet PLOS ONE’s publication criteria as it currently stands. Therefore, we invite you to submit a revised version of the manuscript that addresses the points raised during the review process.

Both reviewers indicated that all points have been carefully addressed in the revised version. A residual point that needs to be addressed in the discussion section is the interpretation of reduced behavioral inhibition in changed GPe pathway functioning in individuals with more drinking. I agree with reviewer 2, that without the support of behavioral data in the current data set, this interpretation indeed should be toned down.

We would appreciate receiving your revised manuscript by Apr 04 2020 11:59PM. To enhance the reproducibility of your results, we recommend that if applicable you deposit your laboratory protocols in protocols.io, where a protocol can be assigned its own identifier (DOI) such that it can be cited independently in the future. For instructions see: http://journals.plos.org/plosone/s/submission-guidelines#loc-laboratory-protocols

We look forward to receiving your revised manuscript.

Kind regards,

Tommy Pattij, Ph.D.

Academic Editor

PLOS ONE

Reviewers' comments:

Reviewer's Responses to Questions

**Comments to the Author**

1. If the authors have adequately addressed your comments raised in a previous round of review and you feel that this manuscript is now acceptable for publication, you may indicate that here to bypass the “Comments to the Author” section, enter your conflict of interest statement in the “Confidential to Editor” section, and submit your "Accept" recommendation.

Reviewer #1: All comments have been addressed

Reviewer #2: All comments have been addressed

2. Is the manuscript technically sound, and do the data support the conclusions?

Reviewer #1: Yes

Reviewer #2: Yes

3. Has the statistical analysis been performed appropriately and rigorously? 

Reviewer #1: Yes

Reviewer #2: Yes

4. Have the authors made all data underlying the findings in their manuscript fully available?

Reviewer #1: Yes

Reviewer #2: No

5. Is the manuscript presented in an intelligible fashion and written in standard English?

Reviewer #1: Yes

Reviewer #2: Yes

6. Review Comments to the Author

Reviewer #1: The authors did a good job in addressing most of my concerns. Without behavioral measurements, the data of this study are not supporting the discussion that changes in GPe pathways in individuals with more drinking are associated with reduced behavioral inhibition. The response of the authors that they were testing alcohol-related changes in GPe pathways in a human model should have enabled them to apply a response inhibition task, if that was one of the main hypothesis of the study. This part of the discussion needs to be toned down or clearly noted as a limitation of the study.

Reviewer #2: (No Response)

7. PLOS authors have the option to publish the peer review history of their article (what does this mean?). If published, this will include your full peer review and any attached files.

Reviewer #1: No

Reviewer #2: No

---

## [Author Response · Author response to Decision Letter 1]

25 Feb 2020

We believe that in this revision of discussion our interpretation of reduced behavioral inhibition is softened. As such, we have added alternative mechanisms that may explain our findings throughout the Discussion section. We have also added to the last paragraph of the limitations section (line 452/453) as follows: “Because we do not have behavioral data on response inhibition in the alcohol intoxication state, there are other potential explanations for our findings.”

We hope the manuscript in its revised form merits publication in PLOS ONE.

---

## [Editor Report · Decision Letter 2]

4 Mar 2020

Alcohol Effects on Globus Pallidus Connectivity: Role of Impulsivity and Binge Drinking

PONE-D-19-29169R2

Dear Dr. Momenan,

We are pleased to inform you that your manuscript has been judged scientifically suitable for publication and will be formally accepted for publication once it complies with all outstanding technical requirements.

With kind regards,

Tommy Pattij, Ph.D.

Academic Editor

PLOS ONE
---

## [Editor Report · Acceptance letter]

6 Mar 2020

PONE-D-19-29169R2 

Alcohol Effects on Globus Pallidus Connectivity: Role of Impulsivity and Binge Drinking 

Dear Dr. Momenan:

I am pleased to inform you that your manuscript has been deemed suitable for publication in PLOS ONE. Congratulations! Your manuscript is now with our production department. 

With kind regards,

on behalf of

Dr. Tommy Pattij 

Academic Editor

PLOS ONE